# Lumican, a Multifunctional Cell Instructive Biomarker Proteoglycan Has Novel Roles as a Marker of the Hypercoagulative State of Long Covid Disease

**DOI:** 10.3390/ijms25052825

**Published:** 2024-02-29

**Authors:** Margaret M. Smith, James Melrose

**Affiliations:** 1Raymond Purves Laboratory, Institute of Bone and Joint Research, Kolling Institute of Medical Research, Faculty of Health and Science, University of Sydney, Royal North Shore Hospital, St. Leonards, NSW 2065, Australia; mobsmith@sydney.edu.au; 2Arthropharm Pty Ltd., Bondi Junction, NSW 2022, Australia; 3Graduate School of Biomedical Engineering, Faculty of Engineering, University of New South Wales, Sydney, NSW 2052, Australia

**Keywords:** lumican, biomarker, tissue pathology, MMP inhibitor, anti-tumour

## Abstract

This study has reviewed the many roles of lumican as a biomarker of tissue pathology in health and disease. Lumican is a structure regulatory proteoglycan of collagen-rich tissues, with cell instructive properties through interactions with a number of cell surface receptors in tissue repair, thereby regulating cell proliferation, differentiation, inflammation and the innate and humoral immune systems to combat infection. The exponential increase in publications in the last decade dealing with lumican testify to its role as a pleiotropic biomarker regulatory protein. Recent findings show lumican has novel roles as a biomarker of the hypercoagulative state that occurs in SARS CoV-2 infections; thus, it may also prove useful in the delineation of the complex tissue changes that characterize COVID-19 disease. Lumican may be useful as a prognostic and diagnostic biomarker of long COVID disease and its sequelae.

## 1. Introduction

Lumican (LUM) is a structural and functional component of the extracellular matrix (ECM) and a class II small leucine-rich KS proteoglycan (SLRP). Mouse studies show LUM is widely distributed in interstitial tissues and is a major bone protein [1,2]. LUM displays 38–45% homology with keratocan (KER) and fibromodulin (FMOD), respectively; thus, it is not surprising that these proteoglycans share overlapping functional properties [3,4]. LUM regulates collagen fibrillogenesis of small regularly spaced collagen fibres in the cornea essential for optical clarity and LUM knock-out (KO) mice display severe disruption in ECM organisation and loss of corneal optical properties [4,5,6], although FMOD regulates the assembly of large collagen fibres in the sclera, providing mechanical stability [3]. LUM is expressed in malignant tumours, including esophageal, lung, gastrointestinal, breast, colorectal, and pancreatic cancers [7,8,9,10,11] but, paradoxically, can display tumour-promoting and anti-tumour activity [12] and is useful as a biomarker, with its expression levels correlating with the severity of tumour grade. A number of pathological tissues display LUM whose roles in ECM assembly and organisation contribute to disease processes leading to its usefulness as a diagnostic biomarker and a means of monitoring the effectiveness of prospective therapeutic reparative procedures [13,14,15]. Small leucine-rich repeat proteoglycans (SLRPs) interact with several cell receptors, growth factors, cytokines and chemokines and have roles in the development of pathological tissues affecting cellular proliferation, migration and differentiation, making them relevant biomarkers of tissue pathology.

SLRPs have crucial roles to play in a varied range of biological processes, in the structural refinement of speciality connective tissues regulating collagen fibril spacing and tissue organization, modulating cell growth with vital roles in cell-matrix interactions in cell-signalling pathways that regulate angiogenesis and wound repair and crosstalk between receptors in inflammation and immune regulation. It is hardly surprising that, armed with these functional attributes, that knowledge of LUM has grown exponentially over the last decade and it is now recognized as a widely distributed instructive multifunctional proteoglycan with considerable potential as a biomarker of tissue pathology and repair.

In this study, we propose platelet-rich plasma (PRP) LUM as a potential new biomarker of the hypercoagulative state induced by SARS-CoV-2 and hence as a possible biomarker of long COVID disease.

## 2. Lumican as a Structural Component of the ECM

LUM is a horse-shoe-shaped superhelix arched protein containing β-sheets on its concave surface and α-helices on its convex aspect [16,17]. This boomerang shape accommodates collagen fibrillar attachment in the concave surface, while keratan sulfate (KS) side chains on the convex surface project outwards to interact with components that regulate normal assembly and growth of collagen fibrils and a variety of other cellular processes [17,18]. The structure of lumican is depicted schematically in Figure 1. Animal SLRP gene knockout studies have established differential effects for individual SLRP family members in ECM assembly [19].

LUM is an ECM matrikine that regulates multiple cellular activities and also has anti-MMP activity, anti-tumour activity [20,21,22] and, promotes corneal epithelial wound healing, regulating gene expression maintaining corneal homeostasis. A LUM peptide designed on the 13 C-terminal amino acids YEALRVANEVTLN (LumC13) binds to activin-like kinase-5/TGFβ type 1 receptor (ALK5/TGFBR1) to promote wound healing. LumC13 forms a stable complex with ALK5 and promotes corneal epithelial cell migration and wound healing [23]. LUM is also an MMP-14 inhibitor [24]. LUM peptides inhibit melanoma spread [25] and suppress pancreatic cancer [26]. Lumcorin is a peptide derived from LUM leucine-rich repeat (LRR) domain 9, which has anti-tumour activity [20]. LUM facilitates inter-axonal crosstalk and the organization of cervical corticospinal innervation in neural networks [27] and protects against skeletal muscle loss [28].

## 3. Lumican as a Mediator, Inhibitor and Regulator

SLRPs interact with a diverse range of cell membrane receptors, cytokines, chemokines, and ECM molecules [29,30], regulating mechano-transduction cell proliferation, migration, and differentiation [31,32]. SLRP interaction with growth factors or tyrosine kinase receptors affects cellular behaviour and tumour progression [33,34,35,36]. SLRPs have roles in the regulation of inflammation through interaction with specific innate immune receptors, coreceptors, and adaptor molecules, to promote a switch between pro- and anti-inflammatory cell signaling. This controls whether the inflammation resolves or becomes a chronic condition [37,38]. Lumican interacts with inflammatory cytokines and chemokines such as CXCL1. SLRPs interact with several growth regulatory receptors that control cellular motility and the immune response and can induce crosstalk between these receptors. The interaction of SLRPs with cytokines and growth factors may also block receptor interactions. In cases of tissue stress or injury, circulating, soluble SLRPs and their proteolytic fragments can act as PAMPs (pathogen-associated molecular patterns)/damage-associated molecular patterns (DAMPs) or alarmins detected by TLRs that regulate innate immunity [39]. Lumican interacts with CD14 and activates the toll-like receptor-4 (TLR-4) pattern recognition receptor as part of the innate immune response, promoting phagocytosis of invading bacteria.

## 4. Lumican in Fibrotic Tissue Pathology

Cardiac fibrosis occurs in virtually all forms of cardiac disease leading to adverse ventricular remodeling and heart failure. LUM levels are elevated in experimental and clinical heart failure [40]. Beta-1,4-galactosyltransferase 5 (B4GALT5) is an enzyme that regulates cardiac fibrosis by interaction with LUM, which activates the serine/threonine protein kinase B (Akt)/glycogen synthase kinase-3 (GSK-3)/β-catenin signalling pathway. Cardiac protein functional properties are intricately linked with post-translational glycosylation events. B4GALT5 is highly expressed in cardiac fibrosis accompanied by tissue changes induced by TGFβ1 and activated cardiac fibroblasts [41]. Knock-down of B4GALT5 decreases the conversion of cardiac fibroblast into myofibroblasts with contractile properties and reduces collagen deposition, whereas overexpression of B4GALT5 elevates cardiac fibroblast activation and regulates the Akt/GSK-3β/β-catenin pathway to promote elevated fibrosis during heart failure [42]. Tissue fibrosis is also a prominent feature of hepatocellular carcinoma, which is the most common form of liver cancer, an aggressive, highly malignant cancer with poor prognosis [43,44]. B4GALT5 is highly expressed in hepatic carcinoma [45], catalyzing the biosynthesis of lactosylceramide via the transfer of galactose from UDP-galactose to glucosylceramide, a central intermediate in the biosynthesis of complex sphingolipids and gangliosides. Sphingolipids are important membrane and lipoprotein components with multiple functions in the central nervous system (CNS) in development, cellular recognition, adhesion and neuronal survival in Parkinson’s Disease, acting as second messengers for growth and differentiation factors and cytokines [46]. A large range of cell biological processes are critically modulated by sphingolipids, including cell growth, migration, adhesion, apoptosis, senescence and cellular inflammatory responses in neurodegenerative and metabolic disorders, cancer development, immune regulation, and cardiovascular and skin disorders [47,48]. Proteomics studies demonstrate LUM as a highly expressed protein in hypertrophic obstructive cardiomyopathy [49] and in oral submucous fibrosis [50]. Elevated expression of LUM in keloids contributes to abnormal collagen deposition [51]; however, LUM can also alleviate hypertrophic scarring by suppressing integrin-focal adhesion kinase (FAK) signalling [52].

## 5. Lumican and Lung Pathobiology

LUM is a major component of adult human lung tissues, whereas decorin, biglycan, and FMOD are minor components [53]. Serum LUM levels are elevated in lung inflammation and asthma [38,54,55]. The pulmonary ECM provides cellular instructive cues that direct the assembly of lung tissue to provide mechanical stability and elastic recoil properties essential for physiological lung function [56]. LUM has a pivotal role in the modulation of pathological vascular remodelling in the lung, which can lead to pulmonary arterial hypertension and stiffening of lung tissue. Cell signaling modulated by LUM prevents the activation of phosphorylated Akt, and suppresses pulmonary arterial smooth muscle activity [57]. LUM and the other SLRPs act as danger signals and signal through the innate immune system to counter infections and fine-tune inflammation and regulation of autoimmune diseases [39]. It has been estimated that 45% of COVID-19 survivors develop pulmonary fibrosis, explaining the long-term impaired lung function evident in long COVID disease [58].

## 6. Lumican and Neurodegenerative Disorders

The extensive collagen fibre networks that support and provide functional properties to tensional and weight-bearing connective tissues do not occur in the CNS. Proteoglycans and hyaluronan provide an alternative means for the stabilization of brain tissue; LUM also regulates brain in-fold formation in the developing human neocortex [59]. When brain and spinal cord are subjected to traumatic injury, a range of chondroitin sulfate (CS)-proteoglycans (hyalectans) are upregulated to stabilize the traumatic defect site by assembling a gliotic scar that prevents propagation of the defect in the brain, one of the softest tissues in the human body. Unfortunately, gliotic scars inhibit functional neural recovery; nerve outgrowth does not occur through these scars and the high CS content in these scars inhibits neural regrowth. It has recently been proposed that SLRPs also play an inhibitory role in these gliotic scars, preventing CNS regeneration [60]. LUM has axonal guidance roles in the formation of neural networks in the brain. In situ hybridization of coronal brain sections reveals LUM is expressed exclusively in the lateral cortex and controls inter-axonal crosstalk between corticospinal neural subpopulations to guide longitudinal axonal migration and ensure the establishment of axonal interconnections with their cognate communicating partners to provide precise interconnected functional neural networks [27]. Differential SLRP gene expression in a 3D blood–brain barrier (BBB) model, validated by quantitative reverse transcription polymerase chain reaction (qRT-PCR) and western blotting, has shown that of the SLRPs examined, LUM was most significantly downregulated in endothelial cells by disturbance in fluid flow. Knocking down LUM expression reduces barrier function in this model; however, adding purified LUM into the hydrogel of this 3D BBB model recovered barrier function under fully developed flow conditions. These findings show shear stress profiles effect cell–matrix interaction in BBB endothelial cells, and identify LUM in the maintenance of barrier function and is required to maintain endothelial cell and BBB barrier integrity [61]. Cross-sectional staining of endothelial cells shows substantial peripheral staining of LUM, suggesting interactions with endothelial cell surface receptors. Previous studies have demonstrated the importance of LUM incorporated into the glycocalyx of kidney endothelial cells in the glomerulus [62]. LUM is also widely expressed in the spinal cord and can interact with inflammatory cytokines such as CXCL1 and several growth regulatory receptors that control cellular motility and the immune response.

Alterations in cerebrovascular blood flow in trauma or disturbance of the BBB in stroke results in a disturbance in the endothelial cell tight junctions and a significant downregulation in LUM gene expression [61] modulating the maintenance of the BBB. Elevated LUM levels have been observed in protein deposits in the brain has been noted in schizophrenia [63]. Progressive neuronal impairment in this condition results from pathophysiological dysregulation of protein production, leading to an abnormal brain composition and the promotion of neurodegenerative processes.

## 7. Lumican and Musculoskeletal Disease

Lumican has been proposed as a discriminative biomarker of lumbar intervertebral disc degeneration (IVDD) and low back pain (LBP) [64] and a longitudinal baseline biomarker reflective of the disc space narrowing, vertebral osteophyte formation, inflammatory marker upregulation and increased mechanosensitivity displayed in IVDD, which correlates with elevated LBP levels [65]. An evaluation of the inflammatory profile of herniated IVDs using proteomic and bioinformatics methodologies found elevated LUM levels in IVD tissues correlated with elevated Modic scores and endplate avulsion [66]. Silencing of LUM expression mitigated a tumour necrosis factor (TNF)-α-induced inflammatory response, cell cycle arrest, and cell senescence evident in herniated IVDs. LUM is involved in the apoptosis signal-regulating kinase-1 (mitogen-activated protein kinase 5, ASK1)/p38 cell signalling pathway affecting the nucleus pulposus (NP) cell phenotype through Fas death receptor ligand expression [67].

LUM inhibits most stages of osteoclastogenesis by suppressing Akt activity but does not inhibit MAP kinases, such as c-Jun N-terminal kinase (JNK), p38, and extracellular signal-regulated kinase (ERK). LUM’s osteoprotective properties stem from its ability to simultaneously increase bone formation and decrease bone resorption [68]. LUM effects on joint fibrosis through TGF-β signaling [69], have been identified in tendon [70]. LUM also upregulates phosphorylation of p38 mitogen-activated protein kinase (MAPK) stimulated myogenesis and is an exerkine protecting against muscle wastage [28].

## 8. Lumican and Cancer

LUM is a biomarker of a number of cancers [71] and is positively correlated with development of oesophageal cancer [10], lung adenocarcinoma and squamous cell carcinoma [11], breast [9], chondrosarcoma [34], colorectal carcinoma [7], pancreatic cancer [8], ovarian [34], bone metastasis arising from lung cancer [72], gastric [73], LUM promotes gastric cancer progression via the integrin β1-FAK signalling [74] and colon cancer [75]. In addition to its protumourogenic properties, LUM also has been reported to inhibit tumour development [12] and suppress pancreatic cancer cell growth [26,76] by stimulating the growth of pancreatic cells and inhibition of tumour cell migration [77]. LUM inhibits snail-induced melanoma migration by blocking MMP-14 activity [21,22,78,79,80,81], inhibits B16F1 melanoma cell lung metastasis [82] and also inhibits prostate cancer development [83]. LUM’s anti-tumour activity stems from specific LUM peptides with anti-tumour activity and its ability to act as an MMP inhibitor, interactive properties with α2β1 integrin also inhibit new blood vessel development required for the nutrition of rapidly expanding tumour cell numbers [24,25]. LUM also has cell directive roles in epithelial-to-mesenchymal transition, cellular proliferation, migration, invasion, and adhesion [71] and is a biomarker of medalloblastoma [84], where its levels correspond with the aggressiveness of such tumours [85].

## 9. Lumican and Retinal Homeostasis

As already discussed, LUM is a retinal SLRP with well-known roles in the regulation of corneal collagen fibril organisation essential for corneal optical properties [86]. LUM is differentially expressed in the retina, choroid and sclera along with KER and FMOD [87]. LUM is correlated with retinal degeneration [88] and is fundamentally involved in retinal homeostasis [86,89,90]. The roles of LUM in inherited and acquired eye diseases have been recently reviewed [91]. A single nucleotide polymorphism in the 5′-regulatory region of the LUM gene (*LUM*) is associated with high myopia in some Asian populations but has a protective effect against myopia in a cohort of English patients [91]. LUM has roles in age-related macular degeneration and diabetic retinopathy, which are leading causes of visual impairment [86,89]. Serum LUM levels are pathological biomarkers of pathological fibrocellular change in the vitreoretinal membrane and idiopathic pathological changes in the epiretinal membrane [89].

## 10. Lumican and Peridontal Disease

Periodontal disease occurs from a buildup of bacteria as a deposit known as tartar, which can affect the tooth attachment to the gums and inflammation of the gingiva and may lead to infection of the cementum, anchoring periodontal ligaments and bony attachment of the teeth. Collagen fibril organization is disturbed in these tissues and LUM levels correlate with disease progression [92,93,94], but its levels are also elevated in periodontal regeneration [95,96].

## 11. Lumican, Liver and Kidney Disease

LUM is a prominent component of kidney tissues with roles in fibrotic kidney disease, inflammation and immune responses [13,38,97,98,99,100,101]. Collagens I, II, III, V, VI, VII, and XV are components of the renal interstitium and their deposition is elevated in renal fibrosis [102]. Upregulation of collagen I and III is an early event in renal fibrosis [98,103]. Tissue fibrosis also occurs in hepatocellular carcinoma [43] and has a global impact [44]. LUM interaction with toll-like receptors in innate immune responses in the kidney act as a trigger of renal inflammation. LUM is a hub gene associated with the accumulation of ECM in diabetic nephropathy, a major cause of end-stage renal disease and a diagnostic biomarker of diabetic nephropathy [13]. Interaction of SLRPs with specific innate immune receptors, coreceptors, and adaptor molecules, promotes a switch between pro- and anti-inflammatory signalling, which controls whether the inflammation resolves or becomes a chronic condition [37,38].

## 12. Lumican and Reproductive Processes

LUM has roles in collagenous re-organisation of the uterine cervix in pregnancy and ovulation [104,105]. Downregulation of LUM promotes the development of pre-eclampsia [106], and is differentially localized in the normal and pathological endometrium and in polycystic ovary syndrome [107]. LUM expression in the murine uterus is modulated by estradiol and progesterone in the oestrous cycle [108,109] associated with ECM remodelling in uterine tissues and with degenerative pathological changes; thus, LUM is a biomarker of tissue pathology [110,111] and early stages of ovulation and may represent a therapeutic target in some forms of female infertility [105]. Endometrial LUM levels are significantly elevated in polycystic ovary syndrome disrupting normal tissue assembly and function and excessive matrix deposition in the endometrium [107].

## 13. Lumican and Adipocyte Regulation

LUM modulates adipocyte functional properties and aids in glucose homeostasis in obesity and insulin resistance associated with type II diabetes [112,113,114]. LUM KO in adipocytes leads to decreased lipolysis, improved adipogenesis and insulin sensitivity in human visceral adipocytes [112]. LUM acts in an extracellular signal regulated kinase (ERK)-dependent manner and is a potential therapeutic target for adipose tissue-targeted therapeutics in type 2 diabetes.

## 14. Lumican and Vascular Disease

Serum LUM levels are elevated in coronary artery disease and correlate with disease severity [115]. Circulating LUM levels are associated with carotid atherosclerosis plaque formation in hypertensive patients and is a promising molecular marker for atherosclerosis [116]. Proteomics analysis and mass spectrometry have shown that LUM is one of several proteins consistently associated with mature atherosclerotic plaque formations [117]. LUM is also a useful biomarker of acute aortic dissections [118,119]. Interaction of LUM with growth factors, cytokines, pathogen-associated molecular patterns and cell surface receptors regulate normal physiology but are also operative in disease processes such as inflammation, innate immune responses and wound healing. These processes are also important in plaque development and progression, making LUM a useful atherosclerosis biomarker [120]. Aortic tissues sampled from hypertensive patients at an early stage of chronic renal failure have a specific gene expression profile with genes with roles in collagen fibrillogenesis, SMC migration and elevated proliferation, particularly LUM, which is a prognostic biomarker in such clinical settings [19]. Mass spectrometric analyses of atherosclerotic aortic tissues have demonstrated a selective enhanced deposition of LUM in the intima of the atherosclerosis-prone internal carotid artery compared with the intima of the atherosclerosis-resistant internal thoracic artery. This enhanced deposition of LUM has important implications for the pathogenesis of atherosclerosis [121]. LUM regulates collagen fibrillogenesis in the ischemic and reperfused rat heart [122]. LUM is deposited in lipid-engorged macrophages in atherosclerotic lesions, where it may regulate the macrophage phenotype. The interaction of LUM with α2-integrin on neutrophils and macrophages regulates their chemotactic migration to atherosclerotic plaques [123,124]. Arterial LUM does not contain KS chains so these interactions are mediated by its core protein [125]. The N-terminal tyrosine sulfate residues of LUM would still be available for interactions with cytokines and growth factors [126,127]. A proteomic study on the human carotid artery showed it contained elevated levels of LUM core protein compared to normal carotid tissue [121], consistent with relative expression levels of LUM expression in coronary arterial disease [19,128] LUM accumulates in hypertrophic cardiomyopathy [42] and experimental and clinical heart failure [40] contributing to accelerated arterial stiffening in coronary heart disease [99] and in the ischemic and reperfused rat heart [122]. Serum LUM levels correlate with the severity of coronary artery disease [115]. LUM is increased during heart failure, cardiac fibrosis, and familial hypertrophic cardiomyopathy and is one of four serum biomarkers of abdominal aortic dissections (AAD) and a participant in TGF-β1 signalling involved in the pathogenesis of AAD [129,130]. LUM deficiency in pulmonary arterial smooth muscle cells promotes arterial remodelling and matrix stiffening in pulmonary arterial hypertension [57].

## 15. Lumican in Virally Induced Fibrinaloid Microclots

Fibrin(ogen) amyloid (fibrinaloid) microclots occur in a range of chronic, inflammatory diseases and may also be induced by a variety of substances, including bacterial inflammogens, serum amyloid A, and the S1 spike protein of SARS-CoV-2 [131]. Measurement of these microclots show potential as a diagnostic when used in conjunction with proteomics, imaging flow cytometry and cytokine analyses for the differential clinical diagnosis of the various diseases associated with such microclots [132]. Such microclots occur frequently in long COVID-19 disease [131,133]. Entrapped proteins within microclots, such as antiplasmin-2 and LUM, are potential biomarkers of these conditions.

It is well accepted that clots that block major blood vessels can lead to strokes or heart attacks; however, tiny clots in the body’s small blood vessels can also be dangerous. Autopsies of patients who have died from COVID-19 disease have shown that many patients have developed tiny so-called “microclots” in their lungs, potentially contributing to respiratory failure. Many researchers believe that microclots cause long COVID symptoms by impeding blood and oxygen flow to the body’s organs and tissues. A growing group of researchers believe these microclots may be responsible, at least in part, for the respiratory deficits displayed in long COVID disease. The term ‘microclots’ reported in patients with post-COVID-19 syndrome, however, technically are not clots. The term ‘amyloid fibrin(ogen) particles’ is a more appropriate term to use for these structures. Fibrinogen is cleaved to form insoluble fibrin clots in the early phases of coagulation and clot formation during wound repair so it is understandable how this terminology arose. Some serum proteins such as LUM and antiplasmin-2 have been observed to be components of these fibrinogen particles and can be identified using proteomics and flow cytometry and have been suggested as biomarkers of the conditions where these fibrinogen particles occur.

## 16. Lumican as a Biomarker

Table 1 lists all the conditions for which LUM (measured in body fluids) has been suggested as a biomarker. With a few exceptions, increased LUM is associated with worsening pathology, prognosis and outcome, especially when measured in serum or plasma.

We could find no other published data on LUM levels in plasma-derived pellets or blood cell precipitates other than that reported in the supplementary data of the 2021 Pretorius study [131].

## 17. Future Research on the Modulatory Cell Instructive Properties of Keratan Sulphate

Keratocan is 38% homologous to and its synthesis is regulated by LUM [151]. Corneal keratan sulfate (KS) is attached to Asn in LUM core protein via a complex-type N-linked branched oligosaccharide in four attachment sites [152], not all sites may be occupied at one time. Nonglycanated LUM core protein is widely distributed in the sclera, aorta, cartilage, liver, skeletal muscle, kidney, pancreas, brain, placenta, bone and lung [2,18,70,125,151,153,154]. The nonglycanated LUM form increases with age due to decreased KS synthesis [155]. The presence of KS-free LUM in tissues may have a significant, but poorly understood impact on inflammation and disease [37].

## 18. The Potential Significance of SLRPs Containing Low-Sulphation KS Chains

Low-sulfation LUM and forms containing high-sulfated KS may be present in specific tissue contexts. KER contains low-sulfation KS chains with instructive properties over neural migration in the embryonic cornea [156]. An interesting story is emerging on neurosensory properties of low-sulfation forms of KS-containing polymers [157,158,159]. KS is a functional electrosensory and neuro-instructive molecule and may participate in cell signalling, particularly in electrically sensitive cell types such as the neuron, although all cells are responsive to electrical stimulation to some degree [160,161]. Some mammals and aquatic species contain electroconductive glycan gels containing KS, which act as electrosensors that signal through their neural networks in a process known as electrolocation [162,163,164,165,166,167]. Of all of the classes of glycosaminoglycan, KS has the highest proton detection capability [168]. Proton transfer in biology is the emotive force that directs cellular behaviour and physiological processes such as oxidative phosphorylation by mitochondria, fundamental not only to energy production (ATP) and life but also to how cell signalling instructive networks signal through kinase and phosphatase enzyme systems.

Biopolymers with proton conductive capability are of considerable interest in bio-nanoelectronics and are being investigated in highly innovative new-generation cell directive devices in biomedicine, such as artificial neural synapses [169]. Nano-origami [170] is a term that has been used to describe methods that have been developed in the last few years for the assembly of defined geometric structures using DNA as a scaffolding material [171]. This has been applied in molecular scale nanophotonics and optoelectronics [172,173], encompassing the computational power of memristor neural network technology [174,175]. This offers significant advances in information transmission technology [3,6] by 3D photonic micro-objects (10–20 μm nanoparticles) [170]. KS has unique biological and electronic properties that indicate it may be of potential application. Such applications may potentially be applied to the repair of traumatically damaged neurons.

## 19. The Discriminative Power of the Sulfation Status of the KS Side Chains of KS-SLRPs Is an Untapped Analytical Parameter Yet to Be Fully Utilized in Biomarker Studies

The full potential of the KS chains on lumican as cell directive entities has yet to be fully explored. Thus, despite being historically neglected in terms of its biology compared to the other classes of GAGs, the full potential of KS as a cell instructive entity and functional ECM component is now coming to the fore in exciting, innovative developments [161,176]. It is therefore important in future studies that full characterization of the KS chains on the various molecular forms of lumican present in tissues should be determined in order that a complete picture is obtained of its pathobiology. Monitoring of the sulfation status of the lumican forms in tissues may further improve the sensitivity of detection and discriminative power of lumican as a biomarker of pathological tissues. Reliance on antibody detection methodology to lumican core protein epitopes for detection, while useful for corroborative data, is less discriminative than KS glycobiology data.

## 20. Instructive Variably Sulfated Keratocan and Lumican in Embryonic Neural Guidance

Sensory trigeminal nerve growth cones innervate the cornea in a highly coordinated fashion [177,178]. KERA and LUM both have documented roles in the regulation of axonal migration during the development of the CNS/PNS [156]. KERA, with minimally sulfated KS chains, may be more permissive of nerve migration in early embryonic development of the cornea and associated retinal neuronal and micro blood vessel networks and may have instructive effects on neuron migration during network formation in non-ocular tissues [179]. Ablation of the KERA gene results in subtle structural changes in the organization of the collagenous ECM but does not perturb the expression of other corneal SLRPs [180]. KERA thus plays a unique role in maintaining an appropriate corneal shape to ensure normal vision. LUM-null mice also exhibit altered collagen fibril organization and loss of corneal transparency but KERA-null mice exhibit a less severe corneal phenotype. The actions of LUM and KERA appear to be coupled at the transcriptional level [151]. In addition to their roles in corneal development KERA and LUM also have key roles in spinal cord development [181], both of these proteoglycans have been immunolocalised in spinal tissues using MAb BKS-1 (+) [181,182]. A proteomics study on corneal KS demonstrated its interactivity with members of the Slit-Robbo and Ephrin-Ephrin receptor families and proteins, which regulate Rho GTPase signalling, actin polymerization and depolymerization in neural development and differentiation, which regulate cell shape and migration [183]. KS decorates a number of highly interactive CNS/PNS proteoglycan (PG) core proteins [161] and has neurosensory properties [184]. Clearly, much still has to be learnt on the cell interactive instructive properties of KS and the impact of its properties in tissues [176].

## 21. Pathobiological Information Provided by the KS Chains of PGs Other Than Lumican

Podocalyxin (PODXL), a highly sialylated type I transmembrane KS proteoglycan expressed on the luminal membrane of brain microvascular endothelial cells, is another illustrative example of the discriminative power of the charge status of its KS chains in normal tissues and tumours that is not picked up by antibodies to core protein epitopes. PODXL is expressed in normal tissues such as kidney, heart, breast, brain and pancreas. Pluripotent stem cells synthesise a form of PODXL containing low-sulfated KS chains [185,186,187]. Upregulation of PODXL correlates with tumour progression, invasion, and metastasis; however, the molecular form of PODXL synthesized by tumour cells contains highly sulfated KS chains and not the low-sulfation KS chains produced by normal embryonic stem cells [188]. PODXL functions as a modulator of BBB function, including metabolite transport, permeability, tight junctions, and immune responses under inflammatory conditions [189]. Low-sulfation KSPGs are detected by antibodies such as MAb R10G or 1B4 [185,186,187,190], KSPGs containing highly sulfated KS chains are detected using MAb 5D4 or MZ15 [185,186,187,188,190,191,192]; these sulfate groups on KS chains are key antigenic and functional determinants [193]. Formerly, studies on the glycobiology of KS in tissues were focused on highly sulfated KS chains since the low-sulfation KS antibodies had yet to be developed so the full story of how all KS isoforms regulated tissues could not be told but is now slowly beginning to unfold [160,161,176]. A KS MAb (BKS-1(+)) has also now been developed, which detects a neo-epitope sulfated galactose KS stub epitope generated by keratanase predigestion [181,182]. LUM and KERA contain low-sulfation KS in the rat spinal cord. LUM is also aberrantly glycosylated in aortic valve stenosis [194]. Future studies may tell a similar story for LUM with high- and low-sulfation KS chains and differential effects on LUM’s cell regulatory properties, proving usefulness as a biomarker in normal and pathological tissues. KS expression is elevated in a 250 kDa KSPG produced by microglia/macrophages following spinal cord injury [195,196,197]; however, KS expression is downregulated in the spinal cord in an animal model of autoimmune neuritis modelling human Guillain–Barré syndrome [198]. This is an inflammatory demyelination disease affecting peripheral nerves involving BBB disruption, infiltration of T cells and macrophages and regional demyelination [199,200,201]. Thus KS is both a marker of traumatic spinal cord injury, where its levels are elevated, and of polyneuropathy, where its expression levels are decreased.

## 22. Conclusions

It is now acknowledged that LUM is not only an ECM component but also a regulator of cell function and a biomarker for several fibrotic pathologies. This multifunctionality suggests its specificity as a biomarker for a single condition may be under question. Measuring this protein in very specific locations, such as the platelet fraction of whole blood, may narrow the scope to a specific pathology of interest. Our proposal that LUM is a hypercoagulation biomarker of long COVID-19 disease represents yet another facet of LUM’s complex, diverse pathobiology.

## Figures and Tables

**Figure 1 ijms-25-02825-f001:**
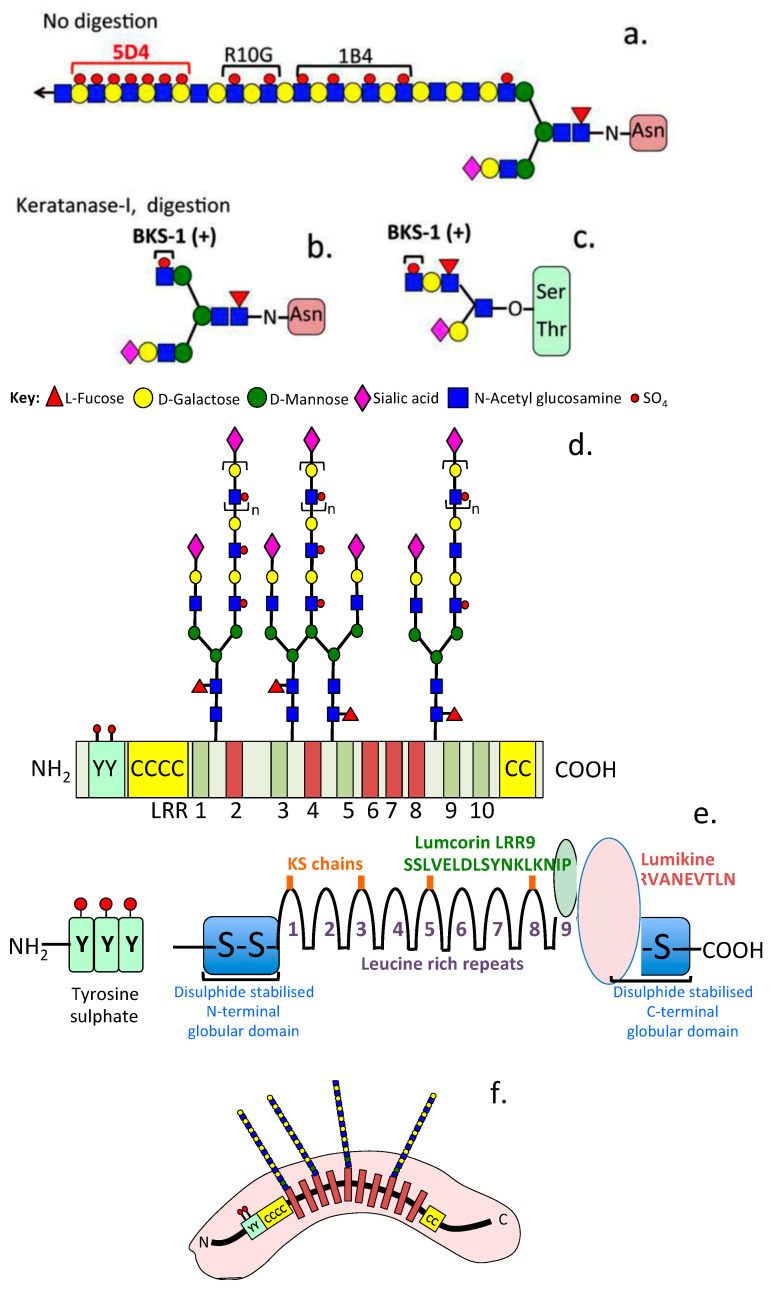
Schematic depiction of the structural organization of lumican. Its KS side chains and antibodies have been used to detect KS. Sulfation along KS side chains is not uniform and contains a highly sulfated region towards the nonreducing terminus detected by MAb 5D4 and regions closer to the reducing terminus that are of lower sulfation detected by MAb’s R10G and 1B4 as depicted (**a**). A neo-epitope MAb (BKS-1 (+)) detects a sulfated reducing terminal GlcNAc epitope in N-linked (**b**) and O-linked KS chains (**c**). Lumican has 10 central leucine-rich repeat modules (LRR), amino and carboxyl disulphide stabilized modules (CC) and an N–terminal region containing sulfated tyrosine residues (YYY). KS chains in lumican are end-capped with sialic acid and are branched structures from the D-Mannose linkage region, Fucose is also a component in the linkage region of KS-I (**d**). Bioactive peptide modules have been identified in the LLR9 domain termed lumcorin and in the carboxyl terminal region of lumican termed lumikine, which is an MMP inhibitor. These regions are schematically depicted by transparent green and red circled regions respectively (**e**). Simplified schematic depicting the KS chains projecting away from the convex surface (**f**). Glycan components are depicted using standard SFNG (symbol nomenclature for glycans) symbols.

**Table 1 ijms-25-02825-t001:** Lumican measured in human fluids and suggested as a potential biomarker.

Condition	Population (n)	Change in Lumican	Reference
In plasma by proteomics and/or ELISA			
Knee osteoarthritis (OA)	Patients (173)	Positively associated with joint space narrowing	[134]
Sepsis	Patients (11) vs. healthy (17)	Higher in sepsis than healthy *p* < 0.01	[135]
Lung adenocarcinoma	Patients (102)	Higher with poorer prognosis	[136]
Metastatic prostate cancer	Patients (62)	Higher with poorer prognosis	[137]
Pancreatic cancer	Patients (40) vs. healthy controls (20)	Higher in cancer	[138]
Bed rest	Healthy males (10)	Decreased with rest	[139]
In serum by proteomics and/or ELISA			
Back pain	OA patients (731)	Higher in group with more pain and inflammation	[64]
IVDD and disc space narrowing	OA patients (723)	Higher with advancing IVDD	[65]
Knee and hip OA	Patients with no (50), moderate (50) or severe OA (50)	Higher in severe OA	[140]
Coronary artery disease	Stable angina pectoris (255)	Higher in advanced disease 0.6 ng/mL vs. 0.4 ng/mL, *p* < 0.001	[115]
Aortic dissection (AD)	Aortic or aneurysm surgery patients (58)	Positively associated with unfavourable p.o. outcome	[141]
Acute aortic dissection (AAD)	AAD patients (14) vs. chronic AD (CAD; 3)	Higher in AAD vs. CAD.	[130]
Acute aortic dissection (AAD)	AAD patients (26) vs. non-AAD (144)	Higher in AAD vs. non-AAD.	[129]
Acute aortic dissection (AAD)	AAD patients (20) vs. healthy (20)	Higher in AAD vs. healthy.	[119]
Acute aortic dissection (AAD)	AAD patients (60) vs. AMI * (30) vs. healthy (30)	Higher in AAD vs. AMI or healthy.	
Carotid artery (CA) atherosclerosis	Patients with (105) or without (71) CA plaque	Higher with CA plaque	[116]
Arterial pressure in obese children	Patients (n = 68)	Positively correlated with higher pressure	[142]
Chronic kidney disease (IgN nephropathy)	Patients (60) vs. controls (43)	Downregulated in advanced disease	[143]
Prostate cancer	Men undergoing radical prostatectomy (557)	In a multivariate model with other biomarkers, predicted recurrence	[144]
Renal cell carcinoma	Patients (99) vs. healthy controls (18)	Positively associated with tumour grade	[145]
Urothelial carcinoma	Patients (30) vs. healthy controls (30)	Higher in carcinoma than healthy *p* < 0.001	[146]
Uterine leiomyoma	Patients (6) vs. healthy (6)	Higher in leiomyoma than healthy	[147]
In amniotic fluid by ELISA			
Preterm birth	Pregancies (252)	Positively associated with inflammation and/or microbial invasion; lower in preterm births	[148]
Preterm birth	Pre-term (36) vs. full-term births (21)	Lower in preterm births	[149]
In aqueous humor by proteomics and ELISA			
Idiopathic epiretinal membrane	Patients (10) vs. age-matched controls (10)	Correlated with central retinal thickness (r = 0.655; *p* = 0.002)	[89]
In CSF by mass spectrometry			
Traumatic brain injury	Patients (16) vs. controls (11)	Negatively associated with favourable outcome	[150]

* AMI = acute myocardial infarction.

## Data Availability

All data is available in the cited publications provided in the bibliography.

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
