# Peer review of "Lumican, a Multifunctional Cell Instructive Biomarker Proteoglycan Has Novel Roles as a Marker of the Hypercoagulative State of Long Covid Disease"

_ijms, 2024, doi:10.3390/ijms25052825_

Round 1

Reviewer 1 Report

Comments and Suggestions for Authors

In this manuscript, the authors present an overview of the roles of lumican as a biomarker of tissue pathology. The topic is described in detail, with many literature references. The paper is interesting, clearly and logically written and can certainly be of interest to readers. I have no substantive comments on this manuscript. In my opinion, it is suitable for publication.

 Nevertheless, the authors should once again check the text from an editorial point of view. For example, the citation style seems to be inconsistent with the IJMS template (there should be [] in the text).

Author Response

The reference entries have been re-formatted IJMS style.

Reviewer 2 Report

Comments and Suggestions for Authors

Manuscript entitled: Lumican, a Pleiotropic Multifunctional Cell Instructive Proteoglycan and Biomarker of Tissue Pathology : Novel Roles as a Prognostic and Diagnostic Biomarker of the Hyper Coagulative State of Long Covid Disease

There are a lot of repeated sentences and ideas in the whole text. Overall, the manuscript is not well organized, and the information in many places is not present with the needed details. Most of the sentences start with “Lumican”. The quality, order and the style of the text must be improved. In the whole text is said that lumican is associated with  something, e.g. disease. It is not clear where is that lumican (blood probably) and what is the association – positive, negative, overexpression, downregulation. This must be improved in the whole text.

Title: Title does not reflect the manuscript content. The second part of title is not needed.

All abbreviations need full text in their first appearance in the text, e.g KS

There are several fuse sentences, e. g, “Lumican regulates collagen fibrillogenesis espe[1]cially small regularly spaced collagen fibres in the cornea essential for optical clarity, lumican KO mice display severe disruption in ECM organisation and loss of optical clarity in the cornea” Page 1

Some sentences require revision: ”Unfortunately gliotic scars inhibit functional neural recovery, nerve outgrowth does not occur through these scars and the high CS content in these scars inhibits neural re-growth.” Page 5

Page 5: “Lumican levels are elevated in lung tissues in adenocarcinoma and squamous cell carcinoma. Lung fibrosis is a prominent feature of COVID-19 disease. It has been estimated that 45% of COVID-19 survivors develop pulmonary fibrosis, explaining the long term impaired lung function in long COVID” The role of lumican is not clear.

Page 6: “Lumican interacts with CD14 and activates the TLR-4 pattern recognition receptor as part of the innate immune response, promoting phagocytosis of invading bacteria.” – reconsider whether the place of this sentence is here.

Page 6: “Lumican is involved in the ASK1/p38 cell signaling pathway effecting the NP cell phenotype through Fas ligand expression” – I do not see what is the connection with the other text in this subsection.

Lumican and cancer – I feel that the statements in this subsection are  controversial. All of the sentences here start with the word lumican.

Page 7: “Lumican is also correlated with retinal degeneration” – how?

Page 7: “Lumican has roles in retinal diseases such as age-related macular degeneration and diabetic retinopathy which are leading causes of visual impairment” – how?

Page 8: “Lumican has roles in the early stages of ovulation …..” – how?

Page 8: “Conversely, treatment with human recombinant lumican increased lipolysis and impaired insulin-sensitivity” – how insulin sensitivity is impaired, when the effect should be favorable?

Page 9: “Measurement of these micro-clots show potential diagnostically when used in conjunction with proteomics, imaging flow cytometry and cytokine analyses for the differential clinical diagnosis of the various diseases associated with such microclot” – I do not understand this sentence

Comments on the Quality of English Language

No specific comments. The text must be revised as it is not well organized. 

Author Response

Reviewer comments

There are a lot of repeated sentences and ideas in the whole text. Overall, the manuscript is not well organized, and the information in many places is not present with the needed details. Most of the sentences start with “Lumican”. The quality, order and the style of the text must be improved. In the whole text is said that lumican is associated with something, e.g. disease. It is not clear where is that lumican (blood probably) and what is the association – positive, negative, overexpression, downregulation. This must be improved in the whole text.

Author response

The manuscript has undergone extensive revision to address these points.

Reviewer comments

Title: Title does not reflect the manuscript content. The second part of title is not needed.

Author response

The title is modified in the revision.

Reviewer comments

All abbreviations need full text in their first appearance in the text, e.g KS

Author response

This has been addressed in the revision

Reviewer comments

There are several fuse sentences, e. g, “Lumican regulates collagen fibrillogenesis espe[1][1]cially small regularly spaced collagen fibres in the cornea essential for optical clarity, lumican KO mice display severe disruption in ECM organisation and loss of optical clarity in the cornea” Page 1Some sentences require revision: ”Unfortunately gliotic scars inhibit functional neural recovery, nerve outgrowth does not occur through these scars and the high CS content in these scars inhibits neural re-growth.” Page 5

Author response

These segments of the manuscript have been modified in the revision.

Reviewer comments

Page 5: “Lumican levels are elevated in lung tissues in adenocarcinoma and squamous cell carcinoma. Lung fibrosis is a prominent feature of COVID-19 disease. It has been estimated that 45% of COVID-19 survivors develop pulmonary fibrosis, explaining the long term impaired lung function in long COVID” The role of lumican is not clear.

Author response

Lumican has well known roles in the regulation of collagenous re-organisation of tissues and in tissue fibrosis which are features of these tissues.

Reviewer comments

Page 6: “Lumican interacts with CD14 and activates the TLR-4 pattern recognition receptor as part of the innate immune response, promoting phagocytosis of invading bacteria.” – reconsider whether the place of this sentence is here.

Author response

This comment has been moved.

Reviewer comments

Page 6: “Lumican is involved in the ASK1/p38 cell signaling pathway effecting the NP cell phenotype through Fas ligand expression” – I do not see what is the connection with the other text in this subsection.

Author response

This is a further example of how apoptosis can potentially be initiated on NP cells in the IVD.

Reviewer comments

Lumican and cancer – I feel that the statements in this subsection are controversial. All of the sentences here start with the word lumican.

Author response

This segment has been reworded. A PubMed search using ‘lumican’ and ’cancer’ uncovered 182 publications. Many of these publications deal specifically with roles for lumican in cancer. Some of these studies are listed below. If roles for lumican in cancer are controversial, then there are an awful lot of researchers who also consider lumican has significant roles to play in cancer so this is not our isolated opinion, see the listed publications below.

Studies which support roles for lumican in cancer.

  1. Cui X, Huo D, Wang Q, Wang Y, Liu X, Zhao K, You Y, Zhang J, Kang C. RUNX1/NPM1/H3K4me3 complex contributes to extracellular matrix remodeling via enhancing FOSL2 transcriptional activation in glioblastoma. Cell Death Dis. 2024 Jan 29;15(1):98.
  2. Gabriele C, Aracri F, Prestagiacomo LE, Rota MA, Alba S, Tradigo G, Guzzi PH, Cuda G, Damiano R, Veltri P, Gaspari M. Development of a predictive model to distinguish prostate cancer from benign prostatic hyperplasia by integrating serum glycoproteomics and clinical variables. Clin Proteomics. 2023 Nov 21;20(1):52.
  3. Guo Z, Li Z, Chen M, Qi X, Sun Z, Wu S, Hou X, Qiu M, Cao Y. Multi-omics analysis reveals the prognostic and tumor micro-environmental value of lumican in multiple cancer types. Front Mol Biosci. 2023 Aug 24;10:1158747.
  4. Berdiaki A, Giatagana EM, Tzanakakis G, Nikitovic D. The Landscape of Small Leucine-Rich Proteoglycan Impact on Cancer Pathogenesis with a Focus on Biglycan and Lumican. Cancers (Basel). 2023 Jul 9;15(14):3549.
  5. Hu G, Xiao Y, Ma C, Wang J, Qian X, Wu X, Zhu F, Sun S, Qian J. Lumican is a potential predictor on the efficacy of concurrent chemoradiotherapy in cervical squamous cell carcinoma. Heliyon. 2023 Jul 8;9(7):e18011.
  6. Gao H, Liu C, Ren Q, Zhang L, Qin W, Wang H, Zhang Y. The Novel SLRP Family Member Lumican Suppresses Pancreatic Cancer Cell Growth. Pancreas. 2023 Jan 1;52(1):e29-e36.
  7. Linke F, Johnson JEC, Kern S, Bennett CD, Lourdusamy A, Lea D, Clifford SC, Merry CLR, Stolnik S, Alexander MR, Peet AC, Scurr DJ, Griffiths RL, Grabowska AM, Kerr ID, Coyle B. Identifying new biomarkers of aggressive Group 3 and SHH medulloblastoma using 3D hydrogel models, single cell RNA sequencing and 3D OrbiSIMS imaging. Acta Neuropathol Commun. 2023 Jan 11;11(1):6.
  8. Zhou Y, Zhou Z, Chan D, Chung PY, Wang Y, Chan ASC, Law S, Lam KH, Tang JCO. The Anticancer Effect of a Novel Quinoline Derivative 91b1 through Downregulation of Lumican. Int J Mol Sci. 2022 Oct 29;23(21):13181.
  9. Chen X, Chen W, Zhao Y, Wang Q, Wang W, Xiang Y, Yuan H, Xie Y, Zhou J. Interplay of Helicobacter pylori, fibroblasts, and cancer cells induces fibroblast activation and serpin E1 expression by cancer cells to promote gastric tumorigenesis. J Transl Med. 2022 Jul 21;20(1):322.
  10. Nizet P, Untereiner V, Sockalingum GD, Proult I, Terryn C, Jeanne A, Nannan L, Boulagnon-Rombi C, Sellier C, Rivet R, Ramont L, Brézillon S. Assessment of Ovarian Tumor Growth in Wild-Type and Lumican-Deficient Mice: Insights Using Infrared Spectral Imaging, Histopathology, and Immunohistochemistry. Cancers (Basel). 2021 Nov 26;13(23):5950.
  11. Athanasiou A, Tennstedt P, Wittig A, Huber R, Straub O, Schiess R, Steuber T. A novel serum biomarker quintet reveals added prognostic value when combined with standard clinical parameters in prostate cancer patients by predicting biochemical recurrence and adverse pathology. PLoS One. 2021 Nov 12;16(11):e0259093.
  12. Dauvé J, Belloy N, Rivet R, Etique N, Nizet P, Pietraszek-Gremplewicz K, Karamanou K, Dauchez M, Ramont L, Brézillon S, Baud S. Differential MMP-14 Targeting by Lumican-Derived Peptides Unraveled by In Silico Approach. Cancers (Basel). 2021 Sep 30;13(19):4930.
  13. Giatagana EM, Berdiaki A, Tsatsakis A, Tzanakakis GN, Nikitovic D. Lumican in Carcinogenesis-Revisited. Biomolecules. 2021 Sep 6;11(9):1319.
  14. Espín R, Baiges A, Blommaert E, Herranz C, Roman A, Saez B, Ancochea J, Valenzuela C, Ussetti P, Laporta R, Rodríguez-Portal JA, van Moorsel CHM, van der Vis JJ, Quanjel MJR, Villar-Piqué A, Diaz-Lucena D, Llorens F, Casanova Á, Molina-Molina M, Plass M, Mateo F, Moss J, Pujana MA. Heterogeneity and Cancer-Related Features in Lymphangioleiomyomatosis Cells and Tissue. Mol Cancer Res. 2021 Nov;19(11):1840-1853.
  15. Sahar T, Nigam A, Anjum S, Gupta N, Wajid S. Secretome Profiling and Computational Biology of Human Leiomyoma Samples Unravel Molecular Signatures with Potential for Diagnostic and Therapeutic Interventions. Reprod Sci. 2021 Sep;28(9):2672-2684.
  16. Yamauchi N, Kanke Y, Saito K, Okayama H, Yamada S, Nakajima S, Endo E, Kase K, Yamada L, Nakano H, Matsumoto T, Hanayama H, Watanabe Y, Hayase S, Saito M, Saze Z, Mimura K, Momma T, Oki S, Hashimoto Y, Kono K. Stromal expression of cancer-associated fibroblast-related molecules, versican and lumican, is strongly associated with worse relapse-free and overall survival times in patients with esophageal squamous cell carcinoma. Oncol Lett. 2021 Jun;21(6):445.
  17. Zang Y, Dong Q, Lu Y, Dong K, Wang R, Liang Z. Lumican inhibits immune escape and carcinogenic pathways in colorectal adenocarcinoma. Aging (Albany NY). 2021 Jan 20;13(3):4388-4408.
  18. Chen D, Smith LR, Khandekar G, Patel P, Yu CK, Zhang K, Chen CS, Han L, Wells RG. Distinct effects of different matrix proteoglycans on collagen fibrillogenesis and cell-mediated collagen reorganization. Sci Rep. 2020 Nov 4;10(1):19065.
  19. Papoutsidakis A, Giatagana EM, Berdiaki A, Spyridaki I, Spandidos DA, Tsatsakis A, Tzanakakis GN, Nikitovic D. Lumican mediates HTB94 chondrosarcoma cell growth via an IGF‑IR/Erk1/2 axis. Int J Oncol. 2020 Sep;57(3):791-803.
  20. Chen X, Li X, Hu X, Jiang F, Shen Y, Xu R, Wu L, Wei P, Shen X. LUM Expression and Its Prognostic Significance in Gastric Cancer. Front Oncol. 2020 May 15;10:605.
  21. Karamanou K, Franchi M, Onisto M, Passi A, Vynios DH, Brézillon S. Evaluation of lumican effects on morphology of invading breast cancer cells, expression of integrins and downstream signaling. FEBS J. 2020 Nov;287(22):4862-4880.
  22. Hsiao KC, Chu PY, Chang GC, Liu KJ. Elevated Expression of Lumican in Lung Cancer Cells Promotes Bone Metastasis through an Autocrine Regulatory Mechanism. Cancers (Basel). 2020 Jan 17;12(1):233.
  23. Salcher S, Spoden G, Huber JM, Golderer G, Lindner H, Ausserlechner MJ, Kiechl-Kohlendorfer U, Geiger K, Obexer P. Repaglinide Silences the FOXO3/Lumican Axis and Represses the Associated Metastatic Potential of Neuronal Cancer Cells. Cells. 2019 Dec 18;9(1):1.
  24. Yang CT, Hsu PC, Chow SE. Downregulation of lumican enhanced mitotic defects and aneuploidy in lung cancer cells. Cell Cycle. 2020 Jan;19(1):97-108.
  25. Sarcar B, Li X, Fleming JB. Hypoxia-Induced Autophagy Degrades Stromal Lumican into Tumor Microenvironment of Pancreatic Ductal Adenocarcinoma: A Mini-Review. J Cancer Treatment Diagn. 2019;3(1):22-27.
  26. Karamanou K, Franchi M, Vynios D, Brézillon S. Epithelial-to-mesenchymal transition and invadopodia markers in breast cancer: Lumican a key regulator. Semin Cancer Biol. 2020 May;62:125-133.
  27. Mao W, Luo M, Huang X, Wang Q, Fan J, Gao L, Zhang Y, Geng J. Knockdown of Lumican Inhibits Proliferation and Migration of Bladder Cancer. Transl Oncol. 2019 Aug;12(8):1072-1078.
  28. Appunni S, Anand V, Khandelwal M, Gupta N, Rubens M, Sharma A. Small Leucine Rich Proteoglycans (decorin, biglycan and lumican) in cancer. Clin Chim Acta. 2019 Apr;491:1-7.
  29. Smith A, Galli M, Piga I, Denti V, Stella M, Chinello C, Fusco N, Leni D, Manzoni M, Roversi G, Garancini M, Pincelli AI, Cimino V, Capitoli G, Magni F, Pagni F. Molecular signatures of medullary thyroid carcinoma by matrix-assisted laser desorption/ionisation mass spectrometry imaging. J Proteomics. 2019 Jan 16;191:114-123.
  30. Yang CT, Li JM, Chu WK, Chow SE. Downregulation of lumican accelerates lung cancer cell invasion through p120 catenin. Cell Death Dis. 2018 Apr 1;9(4):414.
  31. Pietraszek-Gremplewicz K, Karamanou K, Niang A, Dauchez M, Belloy N, Maquart FX, Baud S, Brézillon S. Small leucine-rich proteoglycans and matrix metalloproteinase-14: Key partners? Matrix Biol. 2019 Jan;75-76:271-285.
  32. Chen L, Zhang Y, Zuo Y, Ma F, Song H. Lumican expression in gastric cancer and its association with biological behavior and prognosis. Oncol Lett. 2017 Nov;14(5):5235-5240.
  33. Klejewski A, Sterzyńska K, Wojtowicz K, Świerczewska M, Partyka M, Brązert M, Nowicki M, Zabel M, Januchowski R. The significance of lumican expression in ovarian cancer drug-resistant cell lines. Oncotarget. 2017 Aug 10;8(43):74466-74478.
  34. Jeanne A, Untereiner V, Perreau C, Proult I, Gobinet C, Boulagnon-Rombi C, Terryn C, Martiny L, Brézillon S, Dedieu S. Lumican delays melanoma growth in mice and drives tumor molecular assembly as well as response to matrix-targeted TAX2 therapeutic peptide. Sci Rep. 2017 Aug 9;7(1):7700.
  35. Wang X, Zhou Q, Yu Z, Wu X, Chen X, Li J, Li C, Yan M, Zhu Z, Liu B, Su L. Cancer-associated fibroblast-derived Lumican promotes gastric cancer progression via the integrin β1-FAK signaling pathway. Int J Cancer. 2017 Sep 1;141(5):998-1010.
  36. Appunni S, Anand V, Khandelwal M, Seth A, Mathur S, Sharma A. Altered expression of small leucine-rich proteoglycans (Decorin, Biglycan and Lumican): Plausible diagnostic marker in urothelial carcinoma of bladder. Tumour Biol. 2017 May;39(5):1010428317699112.
  37. Stasiak M, Boncela J, Perreau C, Karamanou K, Chatron-Colliet A, Proult I, Przygodzka P, Chakravarti S, Maquart FX, Kowalska MA, Wegrowski Y, Brézillon S. Lumican Inhibits SNAIL-Induced Melanoma Cell Migration Specifically by Blocking MMP-14 Activity. PLoS One. 2016 Mar 1;11(3):e0150226.
  38. Tsidulko AY, Matskova L, Astakhova LA, Ernberg I, Grigorieva EV. Proteoglycan expression correlates with the phenotype of malignant and non-malignant EBV-positive B-cell lines. Oncotarget. 2015 Dec 22;6(41):43529-39.
  39. Farace C, Oliver JA, Melguizo C, Alvarez P, Bandiera P, Rama AR, Malaguarnera G, Ortiz R, Madeddu R, Prados J. Microenvironmental Modulation of Decorin and Lumican in Temozolomide-Resistant Glioblastoma and Neuroblastoma Cancer Stem-Like Cells. PLoS One. 2015 Jul 31;10(7):e0134111.
  40. Suhovskih AV, Aidagulova SV, Kashuba VI, Grigorieva EV. Proteoglycans as potential microenvironmental biomarkers for colon cancer. Cell Tissue Res. 2015 Sep;361(3):833-44.
  41. Li X, Truty MA, Kang Y, Chopin-Laly X, Zhang R, Roife D, Chatterjee D, Lin E, Thomas RM, Wang H, Katz MH, Fleming JB. Extracellular lumican inhibits pancreatic cancer cell growth and is associated with prolonged survival after surgery. Clin Cancer Res. 2014 Dec 15;20(24):6529-40.
  42. Lieveld M, Bodson E, De Boeck G, Nouman B, Cleton-Jansen AM, Korsching E, Benassi MS, Picci P, Sys G, Poffyn B, Athanasou NA, Hogendoorn PC, Forsyth RG. Gene expression profiling of giant cell tumor of bone reveals downregulation of extracellular matrix components decorin and lumican associated with lung metastasis. Virchows Arch. 2014 Dec;465(6):703-13.
  43. Pietraszek K, Brézillon S, Perreau C, Malicka-BÅ‚aszkiewicz M, Maquart FX, Wegrowski Y. Lumican - derived peptides inhibit melanoma cell growth and migration. PLoS One. 2013 Oct 2;8(10):e76232.
  44. Brézillon S, Pietraszek K, Maquart FX, Wegrowski Y. Lumican effects in the control of tumour progression and their links with metalloproteinases and integrins. FEBS J. 2013 May;280(10):2369-81.
  45. Coulson-Thomas VJ, Coulson-Thomas YM, Gesteira TF, Andrade de Paula CA, Carneiro CR, Ortiz V, Toma L, Kao WW, Nader HB. Lumican expression, localization and antitumor activity in prostate cancer. Exp Cell Res. 2013 Apr 15;319(7):967-81.

Reviewer comments

Page 7: “Lumican is also correlated with retinal degeneration” – how?

Author response

Lumican has well known roles in the regulation of collagenous re-organisation of tissues, collagen fibrils form structural templates in tissues and it is important they are correctly organised for correct tissue function.

Reviewer comments

Page 7: “Lumican has roles in retinal diseases such as age-related macular degeneration and diabetic retinopathy which are leading causes of visual impairment” – how?

Author response

LUM can effect the structural organisation of tissue components in retinal tissues and in blood vessels in diabetic retinopathy causing retinal swelling, changes in epiretinal membranes can also distort vision. Macular degeneration leading to ECM deterioration and blurred vision and ECM changes in retinosa pigmentosa can also contribute to visual impairment.

Reviewer comments

Page 8: “Lumican has roles in the early stages of ovulation …..” – how?

Author response

The early stages of ovulation involve extensiveisECM reorganisation, collagen is a major component of the ECM re-organisation and its assembly is controlled by lumican ensuring correct tissue function in various contexts.

Reviewer comments

Page 8: “Conversely, treatment with human recombinant lumican increased lipolysis and impaired insulin-sensitivity” – how insulin sensitivity is impaired, when the effect should be favorable?

Author response

This segment has been amended to

LUM KO in adipocytes leads to decreased lipolysis, improved adipogenesis and beneficial changes in insulin sensitivity in human visceral adipocytes [123]. LUM acts in an extracellular signal regulated kinase (ERK)-dependent manner and is a potential therapeutic target for adipose tissue-targeted therapeutics in type 2 diabetes.

Reviewer comments

Page 9: “Measurement of these micro-clots show potential diagnostically when used in conjunction with proteomics, imaging flow cytometry and cytokine analyses for the differential clinical diagnosis of the various diseases associated with such microclot” – I do not understand this sentence.

Author response

The following segment has been added to the revised manuscript to better explain this segment.

It is well accepted that clots that block major blood vessels can lead to strokes or heart attacks however tiny clots in the body’s small blood vessels can also be dangerous. Autopsies of patient who have died from COVID-19 disease have shown that many patients have developed tiny so-called “microclots” in their lungs, potentially contributing to respiratory failure. Many researchers believe that microclots cause Long COVID symptoms by impeding blood and oxygen flow to the body’s organs and tissues.

A growing group of researchers believe these microclots may be responsible, at least in part, for the respiratory deficits displayed in Long COVID disease. The term 'microclots' reported in patients with post-COVID-19 syndrome however technically are not clots, the term 'amyloid fibrin(ogen) particles' is a more appropriate term to use for these structures. Fibrinogen is cleaved to form insoluble fibrin clots in the early phases of coagulation and clot formation during wound repair so it is understandable how this terminology arose. Some serum proteins such as LUM and antiplasmin-2 have been observed to be components of these fibrinogen particles and can be identified using proteomics and flow cytometry and have been suggested as biomarkers of the conditions where these fibrinogen particles occur.

Round 2

Reviewer 2 Report

Comments and Suggestions for Authors

No specific comments